# The Efficacy of Sunitinib Treatment of Renal Cancer Cells Is Associated with the Protein PHAX In Vitro

**DOI:** 10.3390/biology9040074

**Published:** 2020-04-07

**Authors:** Rafia S. Al-Lamki, Nicholas J. Hudson, John R. Bradley, Anne Y. Warren, Tim Eisen, Sarah J. Welsh, Antony C. P. Riddick, Fiach C. O’Mahony, Arran Turnbull, Thomas Powles, Antonio Reverter, David J. Harrison, Grant D. Stewart

**Affiliations:** 1Department of Medicine, NIHR Cambridge Biomedical Research Centre, University of Cambridge, Cambridge CB2 0QQ, UK; rsma2@cam.ac.uk (R.S.A.-L.); jrb1000@cam.ac.uk (J.R.B.); 2School of Agriculture and Food Sciences, University of Queensland, Gatton, QLD 4343, Australia; n.hudson@uq.edu.au; 3Cambridge University Hospitals NHS Foundation Trust, Cambridge CB2 0QQ, UK; ayw23@cam.ac.uk (A.Y.W.); tgqe2@cam.ac.uk (T.E.); sarah.welsh@addenbrookes.nhs.uk (S.J.W.); antony.riddick@addenbrookes.nhs.uk (A.C.P.R.); 4Department of Oncology, University of Cambridge, Cambridge CB2 0QQ, UK; 5Scottish Collaboration on Translational Research into Renal Cell Cancer (SCOTRRCC); fiach.o'mahony@ed.ac.uk (F.C.O.); a.turnbull@ed.ac.uk (A.T.); david.harrison@st-andrews.ac.uk (D.J.H.); 6Bart’s Cancer Institute, Charterhouse Square, London EC1M 6BE, UK; Thomas.Powles@bartshealth.nhs.uk; 7CSIRO Agriculture and Food, Queensland Bioscience Precinct, St. Lucia, QLD 4067, Australia; Tony.Reverter-Gomez@csiro.au; 8School of Medicine, University of St. Andrews, St. Andrews KY16 9TF, UK; 9Department of Surgery, University of Cambridge, Cambridge CB2 0QQ, UK

**Keywords:** renal cancer, kidney cancer, sunitinib, PHAX, organ culture

## Abstract

Anti-angiogenic agents, such as the multi-tyrosine kinase inhibitor sunitinib, are key first line therapies for metastatic clear cell renal cell carcinoma (ccRCC), but their mechanism of action is not fully understood. Here, we take steps towards validating a computational prediction based on differential transcriptome network analysis that phosphorylated adapter RNA export protein (PHAX) is associated with sunitinib drug treatment. The regulatory impact factor differential network algorithm run on patient tissue samples suggests PHAX is likely an important regulator through changes in genome-wide network connectivity. Immunofluorescence staining of patient tumours showed strong localisation of PHAX to the microvasculature consistent with the anti-angiogenic effect of sunitinib. In normal kidney tissue, PHAX protein abundance was low but increased with tumour grade (G1 vs. G3/4; *p* < 0.01), consistent with a possible role in cancer progression. In organ culture, ccRCC cells had higher levels of PHAX protein expression than normal kidney cells, and sunitinib increased PHAX protein expression in a dose dependent manner (untreated vs. 100 µM; *p* < 0.05). PHAX knockdown in a ccRCC organ culture model impacted the ability of sunitinib to cause cancer cell death (*p* < 0.0001 untreated vs. treated), suggesting a role for PHAX in mediating the efficacy of sunitinib.

## 1. Introduction

Renal cell cancer (RCC) is the most lethal urological malignancy and the most common type of kidney cancer in adults [1]. Clear cell RCC (ccRCC) is the most common histological subtype, comprising 85% of all cases. The underlying genetic aberrations of ccRCC are becoming increasingly well understood [2]. Von Hippel Lindau (VHL) mutation is the key driver mutation in >90% of ccRCCs, and hence targeting this axis was thought to exploit the disease’s “Achilles heel.” As such, metastatic ccRCC management has been dominated by treatment with anti-angiogenic agents, such as the multi-tyrosine kinase inhibitor sunitinib [3]. Although 70% of patients show a therapeutic tumour response to sunitinib, all patients eventually develop acquired drug resistance. Sunitinib’s mechanism is via endothelial targeting [4], but the impact on RCC epithelial cells, the variable efficacy between patients, and the development of tumour resistance are incompletely understood [5].

One approach for developing a molecular understanding of a disease state and its interaction with drug treatments is to make use of genome-wide transcriptome screening methods. These approaches can simultaneously quantify the entire cohort of mRNA (and other classes of RNA) expressed in a given tissue in a particular set of biological circumstances which makes them very powerful in pathway quantitation. By way of contrast, proteomic approaches provide only limited coverage out of the potentially ≈1 million total human proteins in a complex tissue sample [6,7]. This is in part due to issues with sensitivity, handling an enormous dynamic range and biological interpretation that is complicated by method of acquisition and the particular data analysis platform [8]. On the other hand, transcriptomic data are truly genome-wide and therefore uniquely well positioned to underpin basic molecular discovery in an unbiased manner. The foundational analytical approach when exploring transcriptome data is to compare mRNA levels with each other across two or more treatments, producing a list of differentially expressed (DE) genes. This DE list can then be interrogated in the context of known biological pathways and processes, using functional enrichment web tools such as Gorilla [9] and DAVID [10].

The basic DE approach has limitations because many proteins are differentially activated irrespective of any change in their mRNA levels. Consequently, a reliance on mRNA DE may overlook those genes which behave very differently across states at the protein level, due to one of any number of post-transcriptional mechanisms that could be at play [11]. Here, we have made use of a differential network algorithm called regulatory impact factor (RIF) analysis, which asks the following question of the transcriptome data: “Following sunitinib treatment, which molecule changes its position in the RCC regulatory network the most, irrespective of its own mRNA expression level?” In this context, we find the answer to this question to be the mRNA encoding the protein PHAX. The role of the PHAX protein is subsequently investigated in real patient samples and a tractable in vitro model of RCC. Our data supports an association between sunitinib efficacy and PHAX in vitro.

## 2. Results

### 2.1. Regulatory Impact Factor Analysis

It is evident from examination of the rocket plot or minus-average (MA) plot (Figure 1A) that the mRNA encoding PHAX is neither abundant nor differentially expressed across the two treatments and would not be identified as being of functional interest based on an exclusive reliance on DE. RIF1 and RIF2 scores were plotted and manually explored for outlier mRNA molecules (Figure 1B). Most of the mass of the data are centred close to 0, implying that the majority of molecules are not differentially networked between sunitinib treated and control samples. Most molecules would therefore be predicted to perform the same respective functions in the same manner across the two groups.

However, there are a small number of molecules that are highly differentially connected (based on global patterns of high differential co-expression across the two treatments) according to both versions of RIF. Of the annotated probes, PHAX received the highest combined RIF score (Table 1; Appendix A) based on its extreme position in the top left quadrant of the plot. A number of unannotated probes, such as LOC100130441 and LOC641522, also received extreme scores. We elected to focus on experimentally characterising the role of PHAX given an unambiguous annotation of this probe to an encoded protein.

To further interrogate the functional basis of PHAX’s potential involvement in mediating the impact of sunitinib treatment in RCC tissue we evaluated the pairwise relationships between PHAX and the 1279 DE targets under consideration in more detail. For each target gene we calculated both its phenotypic impact factor (PIF, the product of its DE and average abundance when comparing sunitinib treated versus control RCC tissue) and its differential co-expression to PHAX (again, sunitinib treated versus control RCC tissue). We then ranked the products of those two metrics (full data set in Appendix A). Because we used absolute differential co-expression values for the purposes of generating this list, the sign of the product relates to the direction of DE; i.e., a positive value implies higher expression in sunitinib treated samples, and negative implies lower expression in sunitinib treated samples. The extreme negative values enriched for ribosomal proteins (*p* = 3.92 × 10^−17^; FDR Q value = 2.54 × 10^−13^) including, but not limited to *RPL8*, *RPL23*, *RPL38*, *RPL30*, *RPS14,* and *RPS16*. The extreme positive values enriched for small nuclear RNAs (*p* = 0.0000728; FDR Q value= 0.049) including, but not limited to: *RN7SK*, *RN5S9,* and *RNU4-1*. PHAX is known to regulate this class of molecule [12]. In all cases, functional enrichment was assessed by hypergeometric statistics after importing single ranked lists into the GOrilla webtool [9]. This analysis allows more accurate identification of the molecular pathways predicted to be modulated by PHAX during sunitinib treatment and drive the extreme RIF output scores PHAX was awarded.

### 2.2. PHAX Protein Expression Is Increased in High Grade ccRCC Tumours As Compared to Adjacent Normal Kidney (NK) Cells

Prior to further evaluating the functional effects of PHAX on sunitinib activity in ccRCC, we wanted to determine that PHAX was a relevant protein in this malignancy. As such, we examined the PHAX expression in ccRCC and adjacent NK tissue. NK showed negligible PHAX expression (Figure 2A); PHAX was mildly increased in ccRCC grade 1 tumours (* *p* < 0.05) (Figure 2B). In contrast, a moderate signal was detected in ccRCC grade 2 (** *p* < 0.001), which was more pronounced in high-grade tumours (grades 2–4) (*** *p* < 0.0001) (Figure 2C–E). The pattern of staining was mainly cytoplasmic (Figure 2D,E), with nuclear patterns being seen in some tubular epithelial cells (TECs) and within the glomeruli capillary wall (Figure 2C). Staining intensity presented as digital histological score (D-HSCORE) showed a difference in PHAX expression between ccRCC grades and NK (Figure 2F).

### 2.3. Sunitinib Induces PHAX Protein Expression in Tumour Cells and Vascular Endothelial Cells in ccRCC

To evaluate the functional effects of PHAX, we turned to an established model system of human organ culture of ccRCC and NK tissue [13] to gain insight into the effect of sunitinib on PHAX expression. PHAX expression was analysed by immunofluorescence on sections of organ cultures from grade 2 and 3 ccRCC and NK either left UT or treated with increasing doses of sunitinib (25, 50, 100, and 200 µM) and co-stained for CK. ccRCC organ cultures that were UT and treated with low dose sunitinib (25 and 50 µM) showed rare to mildly infrequent expression of PHAX in CK^-positive^ tumour cells (Figure 3A). In contrast, cultures treated with high dose sunitinib (100 and 200 µM) showed a statistically significant increase expression of PHAX in tumour cells (*p* < 0.001). PHAX expression was also seen in CK^-negative^ infiltrating mononuclear cells within tumour and renal parenchyma interstitium in cultures treated with 100 µM sunitinib (Figure 3A). No signal was detected in a negative control when the primary antibody to PHAX was replaced by isotype-specific antisera (Appendix A). Parallel cultures treated with high dose sunitinib (200 µM) and co-immunostained with anti-CD31 showed marked expression of PHAX in CD31^-positive^ vascular endothelial cells (ECs) in some small and large vessels (Figure 3B). Staining for PHAX was also seen in infiltrating mononuclear cells and in CD31^-negative^ tumour cells (Figure 3B). Organ cultures of NK showed a similar pattern of PHAX expression and CK or CD31 but with less intensity and frequency as compared to ccRCC organ cultures (Appendix A). Additionally present in NK organ cultures were some CK^-positive^/PHAX^-negative^ TECs. Images of PHAX expression in CK^-positive^ tumour cells and ^CD31-positive^ ECs were taken in 10 random high-power fields (×40 magnification) and quantified in an unbiased manner, presented as corrected total fluorescence (CTF) intensity (Figure 3C and Appendix A). A significant increase in PHAX expression was evident in both ECs, tumour cells and in TECs in NK with sunitinib exposure in a dose-dependent manner.

### 2.4. Sunitinib Induces Increased Cell Death in Tumour Cells and in Vascular Endothelial Cells in ccRCC Organ Culture

We next analysed the effect of sunitinib on cell death and cell proliferation in organ cultures of ccRCC and NK. UT ccRCC organ cultures contained sparse TUNEL^-positive^ tumour cells and some vascular ECs (4.1 ± 0.2%). A statistically significant increase in cell death was seen among the same cell population in cultures treated with 25 µM (33.7 ± 0.6%) or 50 µM sunitinib (47.5 ± 0.3%), which was even more pronounced in cultures treated with 100 µM (84.1 ± 0.4%) or 200 µM sunitinib (88.5 ± 0.1%). In contrast, low levels of cell proliferation (detected by immunostaining for pH3^S10^) was seen in ccRCC organ cultures treated with low dose sunitinib (25 µM or 50 µM; 3.7 ± 0.1% and 5.2 ± 0.4% respectively) with a slight increase in cultures treated with high dose sunitinib (100 µM and 200 µM; 5.7 ± 0.7% and 8.2 ± 0.4% respectively). A similar effect of sunitinib-induced cell proliferation was observed in NK organ cultures, but the effects were more pronounced in ccRCC organ cultures (Figure 4, quantified in Figure 5A,B). These data are consistent with the interpretation that sunitinib has its effect by inducing cell death in a dose-dependent manner with minimal effect on cell proliferation in ccRCC.

### 2.5. PHAX siRNA Knockdown Attenuates Sunitinib-Induced Cell Death in ccRCC Organ Culture

We next used siRNA to knockdown PHAX to determine its role in mediating sunitinib’s effect on grade 2 ccRCC in organ cultures. Successful siRNA-induced knockdown of PHAX protein was confirmed by immunofluorescence, co-immunostained with CD31 (Figure 6A). Cultures were either left UT or transfected with a cocktail of PHAX siRNA (containing a mixture of three siRNAs targeting human PHAX) or with each individual PHAX siRNA (PHAXs iRNA-A, B, and C) or a scrambled negative control siRNA (NTsiRNA) for 72 h before treatment with 200 µM sunitinib for 1 h at 37 °C. All cultures were then subjected to PHAX immunostaining. As compared to control groups (UT and NTsiRNA) which showed negligible levels of PHAX expression in CD31^positive^ ECs, in some blood vessels, (Figure 6A,a), a strong signal was seen in cultures treated with sunitinib alone mainly localised to tumour cells, CD31^positive^ ECs, and in some infiltrating mononuclear cells (Figure 5A,e). Notably, sunitinib-induced PHAX expression was markedly reduced in cultures pre-treated with a cocktail of PHAXs iRNA (Figure 6d) and with individual siRNA dup--A or B) (Figure 6A,e,f) but not C (Figure 6A,g). Control group (NTsiRNA) treated cultures alone showed minimal expression of PHAX expression similar to UT cultures (Figure 6A,h), while NTsiRNA in combination with sunitinib (Figure 6A,i) demonstrated a similar pattern to cultures treated with sunitinib alone (Figure 6A,c). Positive controls (transfected with FITC-conjugated siRNA) showed a strong signal in some infiltrating cells (Figure 6A,j). Quantification of the mean signal intensity in ccRCC organ cultures transfected with PHAX siRNA vs. UT is presented as corrected total fluorescence (CTF) intensity in Figure 6B.

We next examined the effect of PHAX knockdown on the cell killing effect of sunitinib treatment in ccRCC organ cultures and adjacent NK (Figure 7A). Following siRNA, all cultures were subjected to TUNEL to determine cell death. In contrast to UT and PHAX siRNA-treated ccRCC organ cultures, which showed only a few TUNEL^positive^ tumour cells and vascular ECs (5.0% ± 0.1% and 5.2% ± 0.9%), cultures treated with high dose sunitinib (200 µM) demonstrated a statistically significant increase in TUNEL^positive^ cells (84.7% ± 1.2%) (mainly consisting of tumour cells and vascular ECs). Importantly, cell death induced by sunitinib was significantly attenuated in cultures pre-treated with PHAX siRNA (37.1% ± 0.49%) or PHAXsiRNA-A (36.2% ± 0.8%) and PHAXsiRNA-B (26.6% ± 0.45%) but not PHAXsiRNA-C (83.1% ± 0.9%). Control cultures pre-treated with NTsiRNA showed a similar signal intensity to UT (3.1% ± 0.2%). Similar effects but to a lesser extent were observed in NK organ cultures (quantified in Figure 7B): These data indicate that PHAX is an important molecule in mediating sunitinib effect in ccRCC as its reduced expression renders tumour cells less sensitive to sunitinib-induced cell death.

## 3. Discussion

In line with the observation that the absolute mRNA expression level of a gene does not necessarily determine its importance in a given biological system, we have computationally predicted that the molecule PHAX is “rewired” in ccRCC patients treated with sunitinib compared to untreated patients with a comparable disease profile. Despite a considerable change in apparent network connectivity following sunitinib treatment, the PHAX mRNA is not differentially expressed in response to the treatment. The rewiring we observe implies some combination of altered protein abundance, post-translational modification (such as phosphorylation), or protein behaviour (such as change in cellular localisation) has occurred in PHAX following sunitinib treatment.

To test whether changes in PHAX contribute to sunitinib’s treatment effect in some way we undertook a sequence of experiments. These experiments reinforce an association between PHAX and sunitinib treatment efficacy in kidney cancer cells. We demonstrate that the encoded PHAX protein is expressed in ccRCC in a functionally relevant manner. Using RNAi in a ccRCC organ culture system we have also shown that PHAX protein expression potentiates the efficacy of sunitinib, a key systemic therapy for the treatment of patients with metastatic ccRCC.

The exact molecular mechanism by which PHAX mediates the effect of sunitinib in controlling growth of kidney tumours is unclear, but PHAX’s physical co-localisation with the tumour vasculature suggests a hypothetical link to the drugs known anti-angiogenic properties. Further, we find PHAX has particularly extreme differential co-expression values to both mRNA encoding ribosomal proteins (e.g., RPL8, RPL23, and RPL38) and small nuclear RNA (e.g., RN7SK, RN5S9, and RNU4-1), a class of RNAs that modify the functions of other RNAs, including transfer RNAs. These data imply that PHAX in part exerts its potentiating impact on the efficacy of sunitinib via these two classes of molecule. The latter observation is particularly intriguing, as although PHAX is not a particularly well characterised protein, its function has previously been linked to the biology of small nuclear RNAs through mediation of nuclear export [14]. Further, in general terms, aberrations in the expression of small nucleolar RNA have previously been linked to cancer phenotypes [15].

The analyses presented in this study have a number of compelling aspects that are of diverse utility. We have used and taken steps towards functionally validating in the setting of cancer biology a novel post-genomic systems biology analysis technique called RIF [16,17]. Differential connectivity as defined by RIF allows predictions of molecular interactions that are of potential functional importance to augmenting the more commonly utilised differential expression assessments. Differential connectivity is important because much functional regulation occurs post-translationally, such as ligand binding, co-factor binding, and cellular localisation.

In a different cancer context, a previous comparative analysis of a range of computational approaches for the identification of key transcriptional regulators showed that, while all methods were able to identify breast cancer relevant regulators, only RIF1 and RIF2 identified regulators with direct connections to ER+ breast cancer [18]. In the case of sunitinib treated RCC tissue, an exclusive reliance on DE of the screened mRNA would have completely overlooked the substantial impact we found the encoded PHAX protein has on the phenotype of interest. Based on the differential connectivity results, we utilised an organ culture model system designed to mirror the human situation to evaluate the functional effects of PHAX in vitro; i.e., cultured ccRCC tissue treated with sunitinib.

We have established that the PHAX protein was upregulated in ccRCCs of increasing grade. This implies PHAX protein abundance may have some prognostic value and/or is somehow relevant in the progression of the untreated disease. Then, we demonstrated that PHAX protein expression in ECs, TECs, and tumour cells was increased by sunitinib in the organ cultures in a dose-dependent manner. Finally, by knocking down PHAX using siRNA, we were able to evaluate the interplay between the cell killing effect of sunitinib and the induction of PHAX protein expression. We successfully knocked down PHAX protein and showed that this reduction of expression resulted in a significantly reduced efficacy of sunitinib in the in vitro model. The use of siRNA in a ccRCC organ culture system represents a valuable experimental platform for future manipulations of ccRCC tissue at the molecular level.

We are aware of several limitations of this research. Tumour tissue used in the organ culture studies was donated by patients with localised ccRCC having surgery, as opposed to the metastatic ccRCC samples used in the original clinical trial cohort providing the gene expression results for differential connectivity analysis. Sunitinib activity is most relevant on distant metastatic tumour deposits, as these are the lethal lesions; nonetheless, this class of agent is known to be active on localised ccRCC, and hence the use of this tissue is considered acceptable [19]. Furthermore, we cannot be categorical about the interplay between PHAX modulation and efficacy of sunitinib therapy in patients or the role of PHAX protein expression/behaviour in the development of drug resistance. For example, we do not know whether the observed increased abundance of PHAX protein in higher grade tumours mediates an effect on sunitinib responsiveness. This sort of question would require further evaluation in samples from a clinical trial.

The systemic therapy options for advanced RCC are rapidly evolving, with combination immunotherapy becoming a first-line treatment option in intermediate and poor-risk patients [20]. However, tyrosine kinase inhibitors such as sunitinib are likely to remain important agents for the treatment of metastatic ccRCC, either as monotherapies or parts of combination therapies. The results from this study show that PHAX protein expression enhances the therapeutic effect of sunitinib in novel in vitro models. Methods of modulation of PHAX could be evaluated to determine if improved efficacy of sunitinib can be achieved.

## 4. Methods

### 4.1. Antibodies and Reagents

Rabbit anti-PHAX antibody (catalogue number ab171321), mouse anti-phosphorylated histone H3^S10^ (pH3^S10^) (catalogue number ab14955), and rabbit anti-phosphorylated histone H3^S10^ (pH3^S10^) (catalogue number ab5176) were all from Abcam Biotechnology, Cambridge, UK. Mouse anti-CD31 (PECAM1, 89C2; catalogue number 3528) and Hoechst-33342 were from Thermo Fisher Scientific, Renfrew, UK. Mouse anti-Cytokeratin (catalogue number VP-c420), TUNEL-label (dUTP^−FITC^) (catalogue number 11767291910), and terminal transferase enzyme (TdT) (catalogue number 03333566001) were from Roche Diagnostics Ltd., Burgess Hill, UK. Anti-mouse-Cytokeratin (catalogue number sc-15367), human PHAX siRNA (a pool of 3 targeted-specific 19-25nts siRNAs (cat: sc-106785), individual siRNA duplex components (cat: sc-106785A; sc-106785B and sc-106785C), control siRNA (Fluorescein Conjugate)-A (sc-36869), and control siRNA-A (sc-37007) were all from Santa Cruz Biotechnology, Heidelberg, Germany. Sunitinib malate (PZ0012) was from Sigma-Aldrich Company Ltd., Gillingham, UK, dissolved in dimethyl sulfoxide (DMSO), and stored as aliquots at −20 °C until use.

### 4.2. Tissue Collection

Fresh-frozen primary ccRCC tissue for analysis was obtained from cytoreductive nephrectomy samples of 23 sunitinib-naive patients with metastatic ccRCC (mccRCC) as part of the Scottish Collaboration On Translational Research into Renal Cell Cancer (SCOTRRCC) study (UK CRN ID: 12229) [21]. Fresh-frozen primary tumour tissue was also obtained from 27 patients with mccRCC treated with three cycles of sunitinib (18 weeks) followed by a cytoreductive nephrectomy after 2 weeks off sunitinib as part of the Upfront Sunitinib (SU011248) Therapy Followed by Surgery in Patients with Metastatic Renal Cancer: A Pilot Phase II Study (SuMR; ClinicalTrials.gov identifier: NCT01024205); 23 of these patients had adequate tissue for analysis (see Table 1 in [22] for summary of patient characteristics). Investigations were approved by institutional review boards, and written informed consent was obtained from each patient. Validation and functional experiments using human tissue were performed with the written informed consent of patients and ethics approval (REC reference: 07/Q0108/49). RCC tissue was obtained immediately after surgical excision and classified according to histological cell type and graded according to the International Society of Urologic Pathologists (ISUP) [23]. Paraffin sections from each batch of tissue (n = 20 from each tumour grade and corresponding non-tumour tissue) were stained with H&E, and the diagnosis was verified independently by an experienced uro-pathologist (AYW).

### 4.3. Nucleic Acid Extraction

RNA extraction was carried out using the miRNEasy Kit (Qiagen, Manchester, UK) according to the manufacturer’s protocols.

### 4.4. Gene-Expression Analysis

As previously described [22], mRNA was amplified using the WT-Ovation FFPE System Version 2 (NuGEN, Redwood City, CA, USA), purified using the Qiaquick PCR Purification Kit (Qiagen, Hilden, Germany), biotinylated using the Encore BiotinIL ModuleIL (NuGEN), purified using the minElute Reaction Cleanup Kit (Qiagen), and quantified using a Bioanalyser 2100 with the RNA 6000 Nano Kit (Agilent, Santa Clara, CA, USA). cRNA was then hybridized to Human HT-12v3 expression Beadarrays (Illumina, San Diego, CA, USA) according to the standard protocol for NuGEN-amplified samples. Gene-expression data were read and normalized with the lump package in R using variance stabilization and robust spline normalization. Illumina expression data were annotated based upon ensemble gene annotation (hg19, release 61). Gene-expression data are available via GEO (accession number GSE65615).

### 4.5. Regulatory Impact Factor (RIF) Analysis

In brief, RIF establishes different patterns of gene network connectivity (via global assessments of differential co-expression) across two states (here, sunitinib treated versus untreated RCC tissue samples). Expressed verbally the algorithm asks the following question: “*Which mRNA molecule is cumulatively the most differentially co-expressed with regard to the highly abundant, highly differentially expressed genes*?” RIF1 and RIF2 are two alternative versions of the same analysis. While RIF1 prioritises regulators that are consistently the most differentially co-expressed with the highly abundant and highly DE genes, RIF2 highlights regulators with the most altered abilities to predict the abundance of DE genes. The outputs of both versions are reported here and the combined information has been used to generate a functional prediction. In both cases the abundances and differential expressions of “target” genes are exploited in conjunction with the differential co-expressions of the “regulators” with respect to those “targets”.

RIF1 and RIF2 were computed as described previously [16,17,24,25]. In this case, the experimental contrast was sunitinib treated (S) vs. control (C) untreated samples, which resulted in 1279 DE genes (*p*-value < 0.001). The entire list of 7350 genes with detectable expression across all S and C samples was treated as the list of potential regulator genes and the RIF metrics for each regulator in *r* computed using the following formulae:$$\mathrm{RIF}1_{r}=\frac{1}{n_{\mathrm{DE}}}\sum_{i=1}^{j=n_{DE}}x_{j}\times d_{j}\times {\mathrm{DC}}_{rj}^{2}$$
and
$$\mathrm{RIF}2_{r}=\frac{1}{n_{\mathrm{DE}}}\sum_{i=1}^{j=n_{DE}}\left[{\left(x_{j}^{S}\times r_{rj}^{S}\right)}^{2}-{\left(x_{j}^{C}\times r_{rj}^{C}\right)}^{2}\right]$$
where $n_{\mathrm{DE}}$ represents the number of DE genes (i.e., $n_{\mathrm{DE}}$ = 1279); $x_{j}$ is the average expression of the *j*-th DE gene across all samples; $x_{j}^{S}$ is the average expression of the *j*-th DE gene across the treated (S) samples; $x_{j}^{C}$ is the average expression of the *j*-th DE gene across the control (C) samples; $d_{j}$ is the differential expression of the *j*-th DE gene in the S vs. C contrast computed as $x_{j}^{S}-x_{j}^{C}$; finally, ${\mathrm{DC}}_{rj}^{2}$ is the square of the differential co-expression between the *r*-th regulator and the *j*-th DE gene, and computed from the difference between $r_{rj}^{S}$ and $r_{rj}^{C}$, the correlation co-expression between the *r*-th regulator and the *j*-th DE gene in the S and C samples, respectively.

### 4.6. ccRCC and Adjacent Non-Tumour Kidney (NK) Organ Cultures

Organ cultures were developed from ccRCC samples from both tumour and adjacent normal kidney (NK) tissues from 4 different RCC patients; ≈1.5 × 1 × 0.5 cm were obtained fresh from surgically excised specimens and developed into organ cultures as previously described [13]. In brief, duplicate <1 mm^3^ fragments of tissue from ccRCC and NK were immersed in tissue culture medium (M199 medium containing 10% heat-inactivated fetal calf serum, antibiotics, and 2.2 mM glutamine). Cultures were either left in media alone (untreated controls; UT) or treated with a dose of sunitinib (25, 50, 100, or 200 µM) for 1 h at 37 °C. A dose response curve was initially carried out, and a dose <25 µM was used, as previously recommended in cell line experiments, but no significant response to sunitinib was seen in organ cultures [24,26]. Cultures were harvested, immediately fixed in 4% paraformaldehyde, and processed for paraffin-wax embedding.

### 4.7. Immunohistochemical Staining for PHAX

Paraffin-wax sections of archives tissue of ccRCC grades 1–4 and adjacent NK were immunostained using our previously described protocol [25]. Antigen retrieval of PHAX involved 2 min in a pressure cooker containing 0.01 mol/L sodium citrate buffer, pH 6.0. Rabbit anti-human PHAX was incubated at 1:100 dilution in blocking buffer (10% foetal calf serum in 0.1 mol/L Tris-HCl containing 0.01% Tween-20) overnight at 4 °C. Sections were further incubated at 1:100 with anti-rabbit horseradish peroxidase-conjugated secondary antibody (Dakocytomation Ltd., Ely, UK) for 1 h at room temperature. Antibody binding sites were visualised using 3,3′-diaminobenzidine (DAB) solution (Sigma-Aldrich) containing 0.01% H_2_0_2_, followed by counterstaining in Mayer’s haematoxylin, and viewed using a Nikon Optiphot-2 microscope (Nikon Corporation, Tokyo, Japan). To quantify the intensity of immunostaining, images of DAB-stained sections were imported into ImageJ version 1.46f (NIH, Bethesda, MD, USA), and the average intensity of the entire field was determined using a method previously described [27]. In brief, the digitalised area was submitted to the plug-in “colour deconvolution” using the built-in vector HDAB, where the staining of haematoxylin and DAB was separated into 3 different panels with DAB only image, haematoxylin, and background. From this image, the software calculated the area in mm^2^, the mean, and the median intensity of DAB, ranging from 0 (black) to 255 (total white). The final DAB intensity was calculated according to the formula *f* = 255 − *i*, where *f* = final DAB intensity; *i* = mean DAB intensity obtained from the software; *i* ranges from 0 (zero = deep brown, highest expression), to 255 (total white). When multiple pictures were taken from the same slide, the mean D-HSCORE was calculated on each. A maximal of 3 regions of interest from the main representative areas were used.

### 4.8. Combined Immunofluorescence for PHAX and Cytokeratin or CD31

Sections of ccRCC and NK organ cultures from UT controls and those sunitinib-treated ones (25, 50, 100, and 200 µM) were subjected to combined-immunostaining for PHAX (1:100 dilution) and cytokeratin (CK; epithelial marker) or CD31 (endothelial marker) (1:500 dilution). Antibody binding sites were visualised using secondary antibody-conjugated to Northern Light-^498^ or Northern Light-^557^ (R&D Systems, Oxford, UK) (1:100 dilution) and incubated 1 h at room temperature. Hoechst-33342 (1 μg/mL) was used for nuclei detection. Species-specific antisera were used as negative controls. Slides were dehydrated in ascending series of ethanol, cleared in xylene, and mounted in DePeX mounts (Sigma-Aldrich, 8 Homefield Rd, Haverhill CB9 8QP, UK). Slides were viewed on a Leica TCS-SPE confocal microscope (CLSM) (Leica Microsystems, Milton Keynes, UK), and the image for each fluorophore was acquired sequentially using the same constant acquisition time and settings rather than simultaneously to avoid crosstalk between channels. Images were then processed in Adobe Photoshop CS6 software. The following formula was used to calculate the mean corrected total fluorescence in organ cultures: (CTF) = (integrated density)—(area of selected cell × mean fluorescent of background readings). Fluorescence intensity was measured using image J version 1.4v, and data were transferred to Microsoft Excel and GraphPad Prism v7.0d (LaJolla, CA, USA) for calculations of means ± SEMs.

### 4.9. Cell Death and Cell Proliferation Assays

The effect of sunitinib treatment on cell death in ccRCC and NK organ cultures was assessed using TUNEL as previously described [13,28]. Briefly, UT control and sunitinib-treated organ cultures were incubated with ^FITC^-dUTP-TUNEL label mix containing TdT-enzyme for 45 min at 37 °C. After thorough washes in MilliQ water, sections were incubated for 10 min at room temperature with Hoechst 33342 for nuclei detection (Molecular Probes, Eugene, OR, USA), mounted in Vectashield Antifade Mounting Medium (Vector Laboratories Ltd., Peterborough, UK) and viewed on a CLSM.

The effect of sunitinib on cell proliferation was analysed using antibody to pH3^S10^ (marker of cell proliferation) (1:500 dilution) and antibody binding sites detected using secondary antibody-conjugated to Northern Light-^498^ or Northern Light-^557^ containing Hoechst-33342. Slides were rinsed in PBS and MilliQ water, and mounted before viewing on CLSM.

### 4.10. Transfection with Specific siRNAs in ccRCC Organ Cultures

ccRCC organ cultures were subjected to siRNA gene knockdown targeting the human PHAX gene. In brief, tissue were either left UT or transfected with human PHAX siRNA (pool of 3 target-specific 19-25nt siRNAs) (catalogue number sc-106785; Santa Cruz Biotechnology, Heidelberg, Germany) or with individual siRNAs duplex components of human PHAX siRNA (siRNA-A, siRNA-B, and siRNA-C) (catalogue number sc-106785-A, B, and C) (all used at 300 nM). Penetration efficiency and the -target effect of the siRNAs were determined using control siRNA (FITC-conjugated) (catalogue number sc-36869) and a negative control siRNA (NTsiRNA; a non-gene-specific, “scrambled” siRNA) (catalogue number sc-37007). Organ cultures were transfected using the transfection reagent Dharmacon Duo (catalogue number T-2010-01; GE Healthcare Dharmacon, Chalfont St Giles, UK) according to the manufacturer’s instructions. Cultures were transfected for 72 h (as this time point provided the optimal knockdown efficiency) followed by treatment with sunitinib for 1 h at 37 °C. Cultures were then harvested, fixed in 4% formaldehyde, and processed for wax-embedding. Sections were subjected to multiplexed immunofluorescence for PHAX and CD31 followed by incubation with corresponding secondary antibodies and viewed on CLSM. Cultures incubated with control siRNA (FITC-conjugated) were also immunostained for CD31.

### 4.11. Statistical Analysis

The average number of TUNEL^-positive^ (including both tumour and vascular compartment cells) was counted in 10 random high-power fields of view (×40 magnification) and divided by the total cell numbers to generate the percentage of positive cells. Counts were carried out on TUNEL stained organ cultures from 3 different patients’ samples. Similarly, the numbers of pH3^S10-positive^ cells were counted and divided by the total cell numbers to generate the percentage of positive cells, calculated as *proliferative index*. A bar represents a mean ± SEM. Each experiment was repeated at least three times and the same statistically significant differences between experimental groups were observed in all three independent experiments, although the absolute values varied. To avoid bias, cell count was standardised using the microscope stage. A beginning point was chosen at random and the next point was obtained by moving in a grid pattern across the section. This resulted in data from 10 unbiased and representative fields per section. Multiple comparisons between groups were analysed using one-way analysis of variance, followed by Bonferroni *post hoc* correction. Microsoft Excel 2016 and GraphPad Prism 7.0d software were used for data processing. Statistical significance was assessed by the analysis of variance test and a p-value of <0.05 was considered significant.

## 5. Conclusions

We wished to learn how the therapeutic drug sunitinib mediates its anti-cancer effect in cancer patients. As a first step in building this understanding, we generated genome-wide gene expression data for sunitinib treated versus control tumour samples. We then used a differential network approach called regulatory impact factors to prioritise mRNA that behaves the most differently following drug treatment. This approach clearly highlighted the mRNA encoding the protein PHAX, despite the fact this mRNA is not differentially expressed in this context. We then used a combination of experiments to demonstrate that (a) the PHAX protein is associated with disease progression in clinical samples, and (b) it can mediate the effectiveness of the drug in an in vitro model of the disease. Overall this study is an example of how the biological complexity implicit in drug-disease interactions can begin to be elucidated using computational tools from the post-genomic era.

## Figures and Tables

**Figure 1 biology-09-00074-f001:**
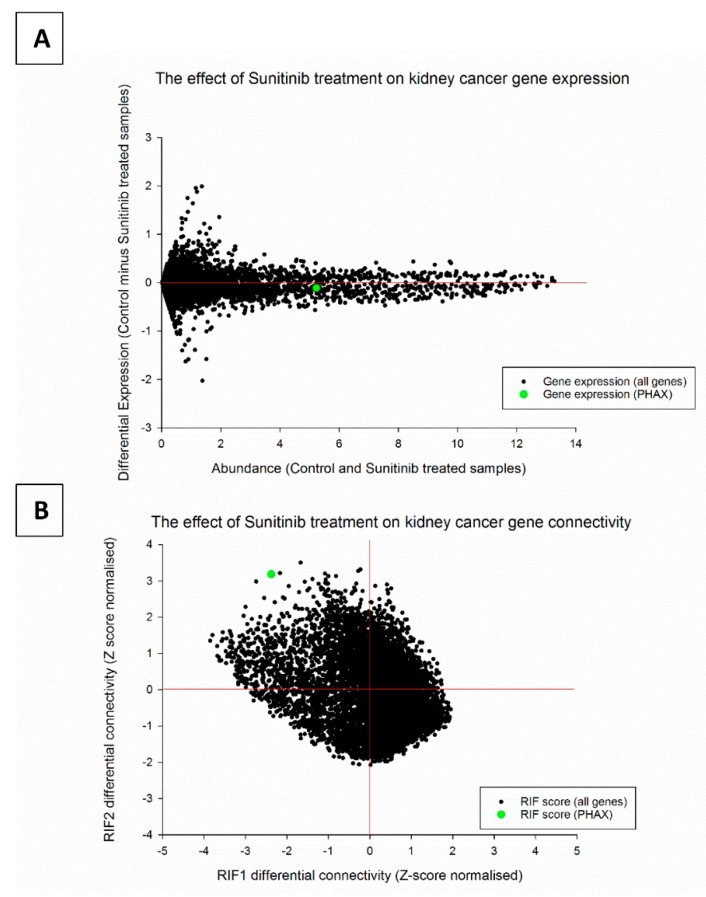
*PHAX* mRNA expression in samples of renal cell cancer (RCC) tissue from patients with metastatic RCC (mRCC) either treated with sunitinib or untreated controls patients. *PHAX* mRNA is moderately abundant and not differentially expressed across the two treatment groups (**A**). However, we found using the regulatory impact factor (RIF) algorithm that *PHAX* mRNA is highly differentially connected between networks constructed using the two groups (**B**). The extreme RIF score for differential networking implies that the encoded protein PHAX behaves very differently in the drug treated versus control samples even though its own mRNA expression level has remained largely unchanged.

**Figure 2 biology-09-00074-f002:**
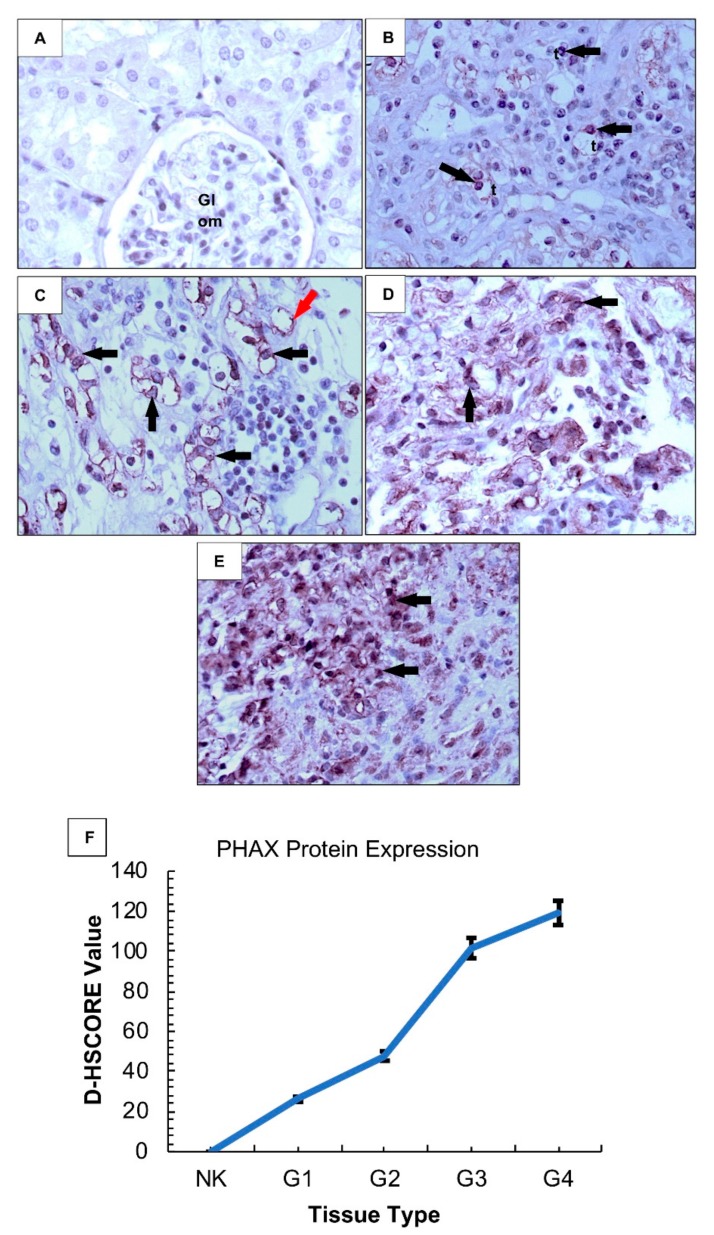
PHAX protein expression in clear cell renal cell carcinoma (ccRCC) grades 1–4 and adjacent normal kidney (NK) cells. (**A**) NK cells show a negligible level of PHAX expression, with a mild signal detected in some tubular epithelial cells (*t*) in ccRCC grade 1 (**B**), increasing in intensity and frequency in high grade tumours (G2–G4) within the intertitial capillary network (*red arrow*) and in sheets of tumour cells (*arrows*), both cytoplasmic and nuclear patternx (**C**–**E**). Signal intensity is presented as a digital histological score (D-HSCORE) with high grade tumours showing statistically significantly increased expression for PHAX (F). (NK vs. G1—(**p* < 0.05), NK vs. G2—(** *p* < 0.01), NK vs. G3 or G4—(*** *p* < 0.0001), G1 vs. G2—(* *p* < 0.05), G1 vs. G3/G4—(** *p* < 0.01), G2 vs. G3—(***p* < 0.01), G3 vs. G4—(ns)). Bars = means + SEMs. N = 3 per group with similar results. Original magnification ×250.

**Figure 3 biology-09-00074-f003:**
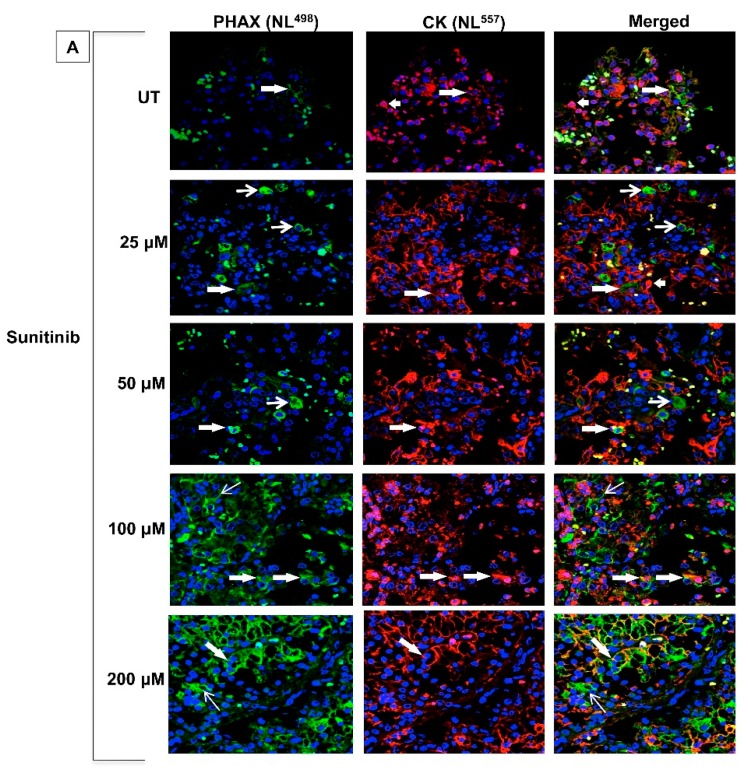
Representative confocal images of PHAX protein expression in organ cultures of ccRCC either untreated or treated with various doses of sunitinib for 1 h in 37 °C. (**A**). Untreated (UT) cultures of ccRCC show a mild and infrequent expression for PHAX is some tumour cells, positive for cytokeratin (CK) (*shaded arrows*). Staining intensity is increased in cultures treated with 25 and 50 µM sunitinib, greatly pronounced in cultures treated with high dose sunitinib (100 or 200 µM). Some infiltrating mononuclear cells within the interstitium are PHAX^-positive^ and CK^-negative^ (*open arrows*), and some CK^-positive^ tumour cells are PHAX^-negative^. (**B**) Parallel sections of 200µM sunitinib-treated ccRCC organ cultures co-stained with antibody to CD31 and PHAX show a marked expression of PHAX in in CD31^positive^ endothelial cells (*arrows*), in some infiltrating cells (*arrowheads*), and in CD31^negative^ tumour cells (*t*). Original magnification ×40. (**C**). Quantification of mean fluorescence intensity calculated as corrected total fluorescence (CTF = integrated density—(area of selected cell × mean fluorescence of background readings) for PHAX/CK co-expression in tumour cells (NK organ cultures (NKoC)—***p* < 0.05—UT vs. 50 or 100 µM; ^+^
*p* < 0.01—UT vs. 200 µM; ^±^
*p* < 0.01—100 vs. 200 µM; RCC organ cultures (RCCoC)—ns (not significant—UT vs. 25 or 50 µM; ^×^
*p* < 0.001—vs. UT, 25 or 50 µM; ^±^
*p* < 0.05—vs. 100 µM). Bars = mean + SEM. N = 3 per group with similar results.

**Figure 4 biology-09-00074-f004:**
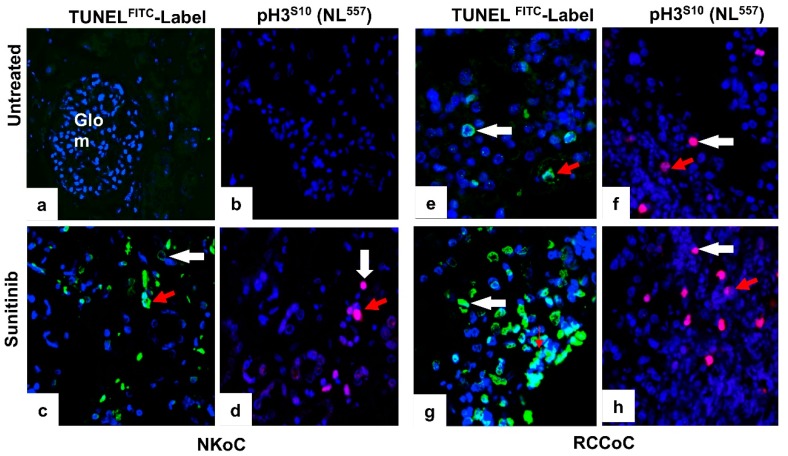
Representative confocal images of TUNEL and pH3^S10^ immunostaining on sections of ccRCC and NK organ cultures either untreated or treated with 200µM sunitinib for 1 h at 37 °C. Untreated (UT) controls of NK (panels **a**,**b**) and ccRCC (**e**,**f**) organ cultures showed negligible signals for TUNEL or pH3^S10^. In contrast, sunitinib treatment resulted in an increased signal for TUNEL signal in NK organ cultures (**c**), more pronounced in ccRCC (**g**). In comparison, sunitinib treatment induced a minimal signal for pH3^S10^ in both organ cultures (**d**,**h**). Tumour cells (*white arrows*), vascular endothelial cells (*red arrows*). N = 3 per group with similar results. Original magnification ×40.

**Figure 5 biology-09-00074-f005:**
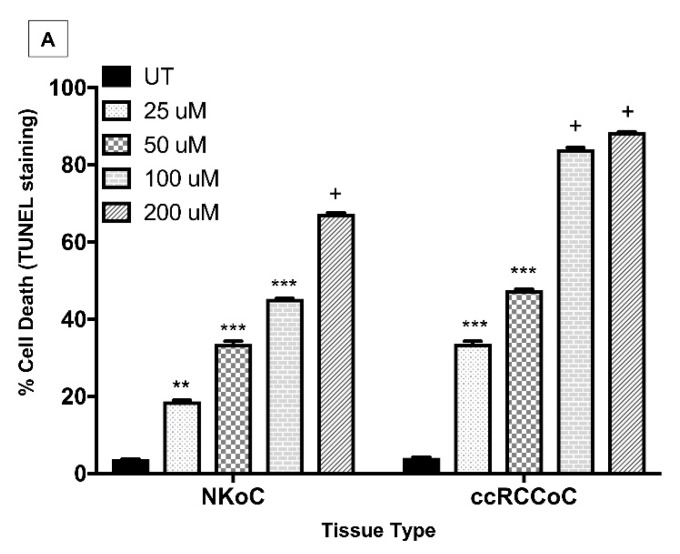
Effects of different doses of sunitinib on cell death (detected by TUNEL) and cell proliferation (detected by anti-phosphorylated histone H3^S10^ immunostaining) in organ cultures of human normal kidney and ccRCC organ cultures. N = 3 with similar results. Statistical significance compared to untreated controls (UT) is indicated as; **p* < 0.05, ***p* < 0.01, ****p* < 0.001, ^+^*p* < 0.0001. NK—***p* < 0.01—UT vs. 25 µM; ****p* < 0.01—UT vs. 50 or 100 µM; ^+^*p* < 0.0001—UT vs. 200 µM. ccRCC—***p* < 0.001—UT vs. 25 µM or 50 µM; ^+^*p* < 0.0001—UT vs. 100 µM or 200 µM. In contrast, sunitinib induced minimal cell proliferation, significantly only at a high dose as compared to UT controls (NKoC-*p* < 0.05 (UT vs. 100 or 200 µM); RCCoC-^+^*p* < 0.05 (UT vs. 50 or 100 µM and ***p* < 0.01-UT vs. 200 µM)).

**Figure 6 biology-09-00074-f006:**
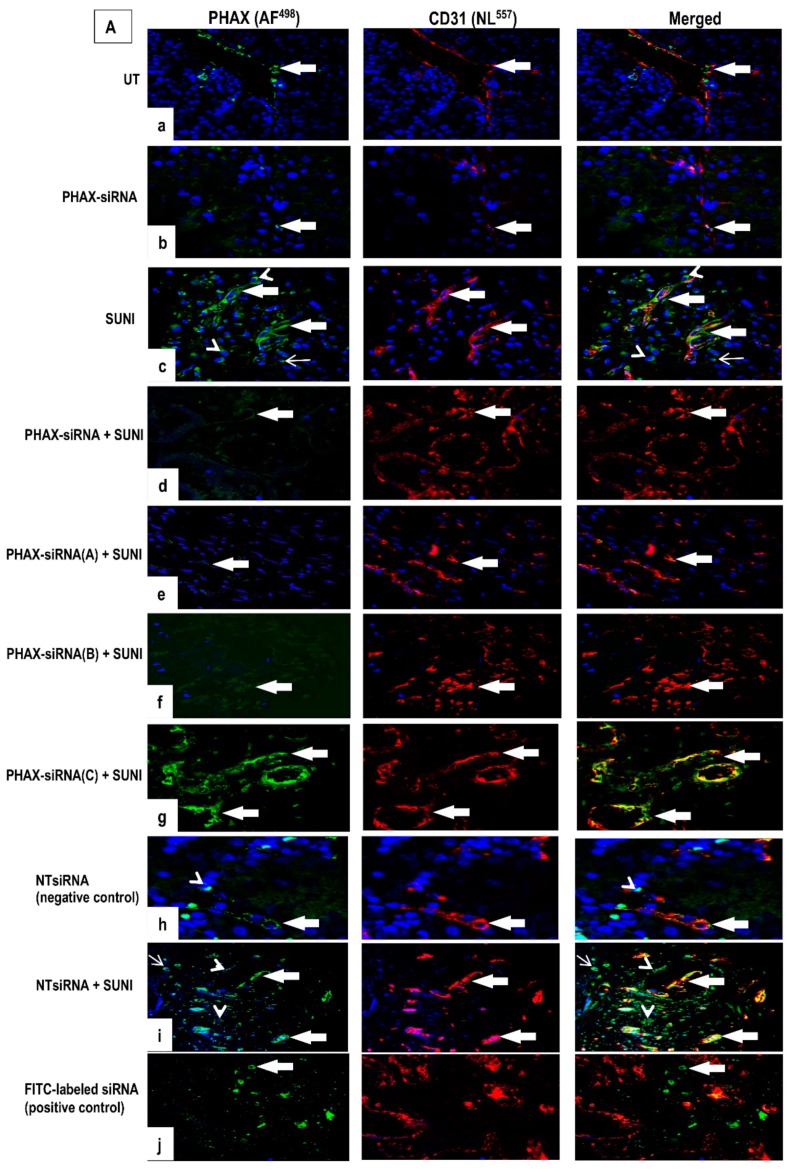
(**A**). Combined immunostaining for PHAX and CD31 in ccRCC organ cultures transfected with a cocktail of PHAX-siRNA; PHAX-siRNA-A, B, C; non-targeting-siRNA (NT-siRNA); and FITC-conjugated siRNA (positive control) prior to treatment with 200 µM sunitinib for 1 h at 37 °C. (**A**). Untreated (UT) cultures and cultures transfected with PHAXsiRNA alone (without sunitinib) show mild, infrequent PHAX expression in endothelial cells (ECs) in some blood vessels (*arrows*) (**a,b**). In contrast, cultures treated with 200 µM sunitinib show a strong signal for PHAX in CD31^positive^ ECs (*arrows*) and in some CD31^negative^ tumour cells and infiltrating cells (*arrowheads*) (**c**). Cultures pre-treated with a cocktail of PHAXsiRNA (pool of three target-specific siRNAs) (**d**) or PHAXsiRNA-A (**e**) or PHAXsiRNA-B (**f**) show a diminished level of PHAX expression, with only a mild infrequent signal seen. PHAXsiRNA-C (**g**) show the same intensity and frequency of staining as cultures treated with sunitinib alone (**c**). Cultures stimulated with NTsiRNA showed a few CD31^negative^ tumour cells (*arrowheads*), and some CD31^positive^ ECs (*arrows*), (**h**) while cultures treated with NTsiRNA followed by sunitinib show a strong signal for PHAX in CD31^positive^ ECs (*shaded arrows*), in infiltrating cells (*open arrows*) and some tumour cells (*arrowheads*) (**i**). Positive controls using FITC-conjugated siRNA showed green fluorescence-labelled cells within the tissue (*arrows*), confirming optimisation of siRNA delivery conditions (**j**). (**B**). Quantification of the mean signal intensity for PHAX protein using Image J calculated as corrected total fluorescence (CTF) in ccRCC organ cultures. Abbreviations: cocktail of PHAX-siRNA, individual siRNAs duplex components of human PHAX siRNA (siRNA-A, siRNA-B, and siRNA-C); non-targeting-siRNA (NT of human PHAX siRNA (siRNA-A, siRNA-B, and siRNA-C)); non-targeting-siRNA (NT-siRNA); and FITC-conjugated siRNA (positive control) prior to treatment with 200 µM sunitinib for 1 h at 37 °C. Image J used to measure fluorescence intensity; data transferred to Microsoft Excel to calculate means + SEMs. Statistical analysis one-way ANOVA and Bonferroni *post hoc* test. *** *p* < 0.0001—vs. UT or PHAXsiRNA or NTsiRNA alone-treated cultures, ^+^
*p* < 0.001—vs. sunitinib-treated cultures; ns—not significant.

**Figure 7 biology-09-00074-f007:**
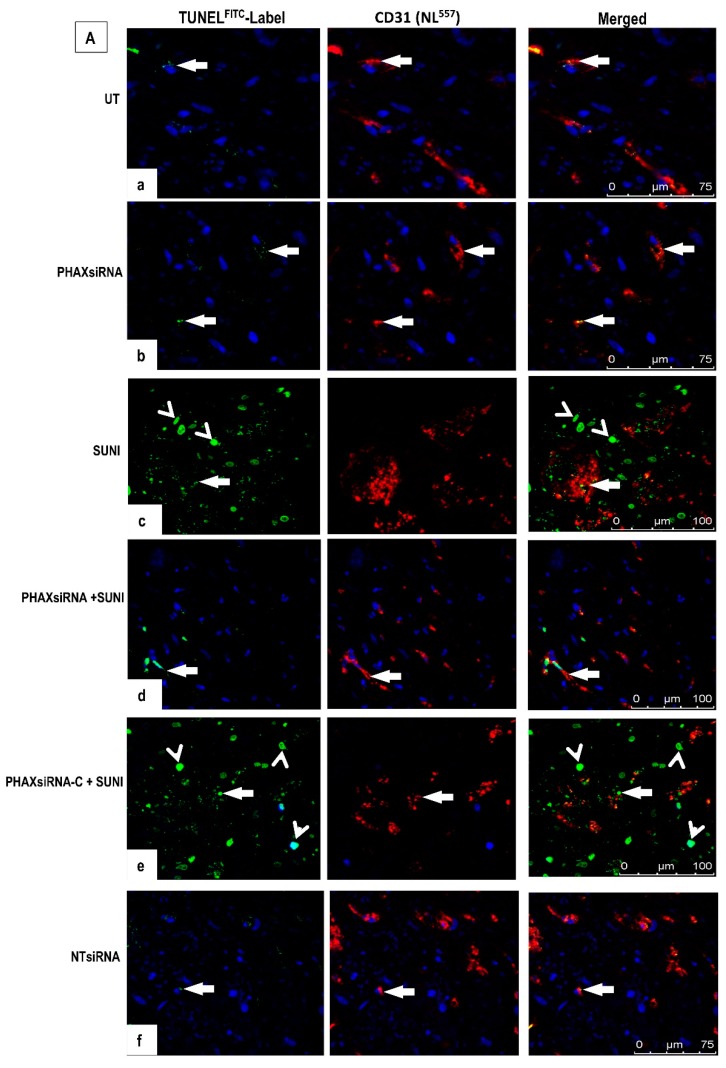
TUNEL^positive^ cells in sections of ccRCC organ cultures subjected to PHAX-siRNA; PHAX-siRNA-A, B, and C; and non-targeting-siRNA (NT-siRNA) prior to treatment with sunitinib (200 µM) for 1 h at 37 °C. (**A**) Untreated cultures and cultures treated with a cocktail of PHAXsiRNA alone show negligible levels of TUNEL ^positive^ cells (*arrows*) (panels **a**,**b**) compared sunitinib-treated cultures, which show increased cell death in some tumour cells (arrowheads) and in CD31^positive^ blood vessel ECs (arrows) (**c**). In contrast, cultures pre-treated with PHAXsiRNA prior to sunitinib treatment show diminished levels of TUNEL^positive^ cells (**d**), while cultures pre-treated with PHAXsIRNA-C prior to sunitinib treatment showed a similar pattern to cultures treated with sunitinib-alone (**e**). Cultures pre-treated with NTsiRNA (negative control) showed a negligible level of staining, similarly to UT cultures (**f**). Original magnification ×40. (**B**). Quantification of the percentage of TUNEL^positive^ cells/total number of cells x100 at ×40 magnification. Statistical significant data are presented as means + SEMs. (*** *p* < 0.0001-UT vs. treated cultures); ns–not significant. One-way ANOVA and Bonferroni *post hoc* test were used to determine statistical significance. *** *p* < 0.0001—vs. UT or PHAXsiRNA or NTsiRNA alone-treated cultures, ^+^
*p* < 0.001—vs. sunitinib-treated cultures; ns—not significant. N = 3 per group with similar results.

**Table 1 biology-09-00074-t001:** The top 10 most differentially connected probes in sunitinib treated versus control kidney cancer cells. Ranking was performed on the absolute average of RIF1 and RIF2 scores. PHAX was awarded the highest combined RIF scores of the annotated probes.

Probe	Gene	RIF1 Score	RIF2 Score	Combined RIF Score
ILMN_3260932	LOC100130441	−2.74	2.98	5.72
ILMN_2190779	PHAX	−2.37	3.18	5.55
ILMN_2141030	LOC641522	−2.16	3.21	5.38
ILMN_2110751	CHRNA5	−3.79	1.50	5.30
ILMN_1787314	ALS2CR14	−2.99	2.28	5.27
ILMN_2169839	CNBP	−3.85	1.34	5.20
ILMN_3179148	LOC100128096	−1.66	3.50	5.17
ILMN_2053536	RHBDL2	−3.54	1.58	5.12
ILMN_3279960	LOC642784	−3.04	2.03	5.08
ILMN_1680774	LOC730994	−2.54	2.52	5.07

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
