# Peer review of "The Efficacy of Sunitinib Treatment of Renal Cancer Cells Is Associated with the Protein PHAX In Vitro"

_biology, 2020, doi:10.3390/biology9040074_

Round 1

Reviewer 1 Report

This study tried to evaluate the interaction between sunitinib efficacy and PHAX protein expression in RCC. The topic is novel but lots of shortcomings needed to be overcome or explained.

Although authors tried to rationalize PHAX as the potential candidate in this study, there were several genes (points) located nearby PHAX in Figure 1B. Albeit authors explained as “A number of unannotated probes, such as LOC100130441 and LOC641522, also received extreme scores (line 101)”, it is not easily to conclude that all genes of extreme position were all unannotated or function unknown protein. The selecting criteria of PHAX remained hard to realize. In Figure 2B-D, there was no distinguishable ccRCC regions showed in representative figures, they were looked like adjacent paratumoral renal parenchyma tissues. Only Figure 2E showed obvious ccRCC features, which made it doubtful that the increasing trend of PHAX among RCC aggressive differentiation was deduced from scores of non-tumoral tissues. In organoid culture model, sunitinib conferred no anti-angiogenic effect on those ccRCC tissues (line 153 “Parallel cultures treated with high dose sunitinib (200μM) and co-immunostained with anti-CD31 showed a marked expression of PHAX in CD31-positive vascular endothelial cells (ECs) in some small and large vessels (Figure 3B). Staining for PHAX was also seen in infiltrating mononuclear cells and in CD31-negative”), did those ccRCC obtained from sunitinib resistant patients? And there is no Figure S1-S3 available on upload files. In Figure 4, sunitinib induced higher apoptosis and inhibited cell proliferation in either NK or ccRCC organoid. It should be reasonable for the function of sunitinib. But in Figure 5 and 6, sunitinib induced PHAX expression and PHAX silence reduced sunitinib induced apoptosis. authors tried to explain that “reduced expression of PHAX renders tumour cells less sensitive to sunitinib-induced cell death”, but it was not easily to explain why sunitinib induced very higher expression of PHAX, it implicated that all sunitinib treated ccRCC eventually became sunitinib-resistant because of sunitinib upregulated PHAX expression. Despite organoid culture was a good strategy for analyzing interaction between cancer and its microenvironment, and it should show related quantitative figures or tables to help this draft more realizable. And it should also be helpful to perform some molecular interaction in cell line experiments such as western blot to investigate more detail regulatory mechanisms of PHAX in sunitinib treatment of RCC. Several typographic and grammatical  mistakes remained, and they need to be corrected and revised more precisely.

Reviewer 2 Report

Dear Authors,

I found the paper to be well-written and the experiments well-designed and presented.

I only found minor errors. Please carefully check all the provided information - eg. Santa Cruz Biotechnology is not located in Heidelberg, but in Dallas, Texas (headquarters). The same for Qiagen (line 382) and other companies.

Also, in line 485 - are you sure that the used siRNA concentration was 600 nM?

Moroever, there should be a space between the numerical values and units - eg. 200 uM, instead of 200uM, etc.

Lines 468 vs 502: SEM vs S.E.M.? Please check.

Also, please consider to replace Table 1 with a graph showing the same data. Maybe it will be easier for readers to follow.

Figure 3c - it is very difficult to analyze the graph due to the very small font and quality of the bars, etc.

Reviewer 3 Report

The authors present an extremely interesting validation of a novel post hoc transcriptomic analysis known as RIF to identify the PHAX protein as important for sunitinib anticancer mechanism of action in renal cell carcinoma.  The authors prove this point with very nice organoid experimentation from patient samples using specific PHAX knockdown and present the data in a very organized and clear format.  Especially well done is the in depth analysis of effect of PHAX on cell type within the organoid system.  It is also interesting to see how few genes are differentially expressed before RIF analysis but how many genes show differential network connectivity after RIF analysis, and suggest the complexity of post transcriptional and post translational modifications.  It would be interesting to see if these genes identified by RIF also show noncanonical functions or changes in posttranslational modifications.  

The authors discussed reasonably well the limitations in this study, in particular the use of nonmetastatic tissues.  There may be another limitation however the authors might want to consider for future studies the diversity in network connectivity among patients within a cohort.

Altogether this is a very interesting paper and no modifications are needed.

Reviewer 4 Report

The manuscript by Al-Lamki at el identified a novel biomarker PHAX for indicating the efficacy of sunitinib treatment of ccRCC. It's an interesting topic with promising clinical significance. However, the results in current version are not sufficient to support their conclusion. Here are some concerns. 

Overall in which cell type PHAX expressed has not been precisely addressed. The authors can not simply define “tumor cells” only because the cells were tested CK+, or CD31- by immunofluorescence. Without other diagnostic ccRCC markers such as CD10/EMA/Vimetin, only CK or CD31, or the combination is not sufficient to make such a statement. 

  1. Fig. 1A, based on observation that PHAX was clearly upregulated in ccRcc tissues and enhanced by sunitinib at protein level, I would speculate a PHAX expression at transcription/mRNA level. Can author argue why there was no remarkable abundance nor differential expression of PHAX in sunitinib treated and untreated groups as shown by MA plot.

  1. Are there any previously identified sunitinib-targeted downstream molecules also pop up in the new designed RIF algorithm. Validation using conventional methods is always required to make the data more convincing.  

  1. Fig. 3A, CK is a widely used maker for demonstrating whether a given tumor is of epithelial origin, however it doesn’t seem to be a good marker for ccRCC. In fact, it has been shown previously positive staining in less than 50% of the metastatic ccRCC paraffin tissues (Cheol Lee, et al). Again, it would be a requirement to include at least one extra ccRCC-specific marker as a positive control or other appropriate negative controls in the exp design.

  1. Fig. 5A, it would be better to perform QPCR or, if there is an antibody available, western blot to confirm the knockdown efficiency to diminish the potential bias arising from the background of staining.

  1. Did author also find a similar trend in their tissue culture system that PHAX was increased in high grade ccRCC tumors? If so, I would expect a more intense signal of PHAX in Fig. 5A-a as the tissue was from Grade 4 ccRCC patients as stated in the context.

  1. There is no Table 1 for summary of patient characteristics as stated in Method 4.2, which should definitely be included in the paper.

  1. According to Method 4.2, 23 sunitinib-naïve patient tissues and 27 patient tissues were included in this study. How many paraffin tissues were tested for PHAX expression? What is the approximate positive ratio of PHAX expression in ccRCC?

9. No supplement data attached.

Round 2

Reviewer 1 Report

It should be acceptable.

Author Response

We would like to thank the reviewer for accepting the revised version of our manuscript.

Reviewer 4 Report

The IF staining in supplementary files is much clearer than those in main figures. Can authors adjust the resolution for those figures? 

Author Response

We thank the reviewer for raising this comment and have now submitted a zip file of high resolution images of Figures 1-7.
